# Cognition and Neuropsychological Changes at Altitude—A Systematic Review of Literature

**DOI:** 10.3390/brainsci12121736

**Published:** 2022-12-19

**Authors:** Kathrin Bliemsrieder, Elisabeth Margarete Weiss, Rainald Fischer, Hermann Brugger, Barbara Sperner-Unterweger, Katharina Hüfner

**Affiliations:** 1University Hospital of Psychiatry II, Department of Psychiatry, Psychotherapy, Psychosomatics and Medical Psychology, Innsbruck Medical University, 6020 Innsbruck, Austria; 2Department of Psychology, University of Innsbruck, 6020 Innsbruck, Austria; 3Department of Pneumology, Medizinische Klinik Innenstadt of the Ludwig-Maximilians-University of Munich, Ziemssenstr. 1, 80336 München, Germany; 4Institute of Mountain Emergency Medicine, Eurac Research, Via Ipazia 2, 39100 Bolzano, Italy

**Keywords:** altitude, hypoxia, neuropsychological tests, cognitive domains, cognition, PRISMA, STAR data reporting guidelines

## Abstract

High-altitude (HA) exposure affects cognitive functions, but studies have found inconsistent results. The aim of this systematic review was to evaluate the effects of HA exposure on cognitive functions in healthy subjects. A structural overview of the applied neuropsychological tests was provided with a classification of superordinate cognitive domains. A literature search was performed using PubMed up to October 2021 according to PRISMA guidelines. Eligibility criteria included a healthy human cohort exposed to altitude in the field (at minimum 2440 m [8000 ft]) or in a hypoxic environment in a laboratory, and an assessment of cognitive domains. The literature search identified 52 studies (29 of these were field studies; altitude range: 2440 m–8848 m [8000–29,029 ft]). Researchers applied 112 different neuropsychological tests. Attentional capacity, concentration, and executive functions were the most frequently studied. In the laboratory, the ratio of altitude-induced impairments (64.7%) was twice as high compared to results showing no change or improved results (35.3%), but altitudes studied were similar in the chamber compared to field studies. In the field, the opposite results were found (66.4 % no change or improvements, 33.6% impairments). Since better acclimatization can be assumed in the field studies, the findings support the hypothesis that sufficient acclimatization has beneficial effects on cognitive functions at HA. However, it also becomes apparent that research in this area would benefit most if a consensus could be reached on a standardized framework of freely available neurocognitive tests.

## 1. Introduction

In the last century, mountaineering has gained in popularity and is now a mass phenomenon with a continually growing number of individuals hiking, skiing, climbing, and trekking in high-altitude (HA) environments. Additionally, HA exposure can occur in work-related contexts (aviation and astronauts or health care services and alpine rescue). However, at higher altitude the reduced partial pressure of oxygen and the decreased barometric pressure can cause acute mountain sickness (AMS) [1], or other possibly life-threatening diseases such as high-altitude pulmonary (HAPE) [2], or cerebral edema (HACE) [3]. With a ratio of about 21 percent of oxygen, the composition of the atmosphere at HA is roughly the same as at sea level. However, the total atmospheric pressure, and therefore oxygen pressure, declines exponentially with altitude [4,5,6]. Symptoms of altitude disease can occur when reaching above an elevation of about 2500 m (8202 ft). HA symptoms become more severe with increasing altitude, and with faster ascent rates, with the decisive factor for acclimatization being the sleeping altitude (of about 1500 m [4921 ft]) [4]. Altitude is classified as medium (1500–3000 m [4921–9843 ft]), high (3000–5500 m [9843–18,045 ft]), or extreme (>5500 m [>18,045 ft]) [6].

Recent reviews focusing on the effects of hypobaric hypoxia on brain functions reported a variety of transient neurological consequences, including HA headache, sleep-disturbances, mental disorders (mood state disturbances, acute psychosis, anxiety) and cognitive impairment [7,8]. Overall, previous research mostly demonstrates that altitude exposure can lead to cognitive impairment. Cognition in these studies has often been seen as a single summary construct and few studies have looked at individual cognitive domains (see also Section 1.1). In addition to hypobaric hypoxia, other physical stressors at HA have also been reviewed concerning their impact on certain cognitive domains, including cold temperatures [9,10], movement strains [11] as well as a lack of recreation and sleep [12]. Polysomnographic studies identified reductions in slow wave sleep as the most consistent altitude-induced change in sleep structure and this decrease in neuronal synchronization during sleep at altitude also appeared to be associated with a decrease in sleep-related memory consolidation in healthy subjects [13]. In addition, mental stressors such as isolation, a new environment, etc., can affect cognition [14,15].

With the brain being more dependent on a constant oxygen supply than many other tissues, there are some brain areas that are especially susceptible to oxygen depletion. To some extent, due to their distal position in the vascular distribution [14], this includes the hippocampus, basal ganglia, and the cerebral cortex [16,17]. The hippocampus plays an important role in learning and memory, as does the cerebral cortex, which is also responsible for higher-level processes of the human brain, such as executive functions [14,18].

In the present review, effects of altitude exposure on cognitive functions in healthy subjects are systematically reviewed and the applied neuropsychological tests are classified according to their superordinate cognitive domains.

### 1.1. Current State of Research

Several reviews relevant to this work have been published in recent years.

One of the older works by Virués-Ortega, Buela-Casal, Garrido, and Alcázar (2004) [19] reviews the effects of moderate, high, and extreme altitude on neuropsychological functioning. A prolongation of reaction time and changes in accuracy and motor speed were noted, the latter already at lower altitudes. Impairments in encoding and short-term memory were observed mainly at altitudes above 6000 m (19,685 ft). Furthermore, deficits in verbal fluency, speech production, cognitive fluency, and metamemory were noted. They also summarized decreased tactile, olfactory, pain, and taste thresholds as well as somesthetic illusions and visual hallucinations.

A work by Petrassi, Hodkinson, Walters, and Gaydos (2012) [20] included 97 publications (among them 75 articles) and examined hypoxic impairments with respect to aviation-related tasks at moderate altitude, such as mental functions as well as sensory deficits. The results of this study revealed deficits in learning, reaction time, decision making, and certain types of memory, although the authors note that the results appear inconsistent and difficult to replicate. The literature on hypoxic visual deterioration, on the other hand, was found to be more consistent and demonstrated such impairment.

The focused review by Taylor, Watkins, Marshall, Dascombe, and Foster (2016) [21] collected evidence for the impact of temperature, such as heat stress and cold stress, as well as hypoxia, on cognitive function. The accumulated evidence of 16 articles concerning hypoxia suggested that the effects of hypoxia on cognitive function are dependent on both task complexity and altitude level.

McMorris, Hale, Barwood, Costello, and Corbet (2017) [22] included 22 experiments for a systematic meta-regression analysis of the effects on central executive and non-executive tasks. They found low p_a_O_2_ (arterial oxygen partial pressure), especially between 46.7 and 80 hPa (35–60 mmHg), to be the main predictor of cognitive performance decline. Interestingly, the effect proved to be independent of whether the exposure was conducted under hypobaric or normobaric hypoxic conditions. Contrary to their expectation, no significant differences were found between central executive and non-executive, such as perception/attention and short-term memory, tasks.

A very detailed systematic review by Martin and colleagues (2019) [10] examined various environmental stressors such as heat, cold, and hypoxic stress on cognitive performance. They included 59 articles on hypoxic stress, with studies that investigated military task performance among them. Although the effects of hypoxia on cognitive performance were heterogeneous, the authors concluded that the severity and duration of the stay at HA had the greatest influence on cognition, rather than the complexity of the tasks. Furthermore, the interindividual variation had a large influence on the results. Last but not least, altitude acclimatization seems to have a positive effect on cognitive performance, but an optimal protocol is still pending.

### 1.2. Critical View upon the State of Research

Comparisons between studies examining cognitive performance under laboratory-induced hypoxia and at HA are hampered by the methodological differences between them, including subject characteristics (normal persons to high end climbers [23,24]; the vast diversity of neuropsychological tests [25,26,27,28,29]; the different study designs (e.g., field expeditions versus controlled environmental chamber studies) [23]; as well as differences in duration and severity of hypoxia, and exposure with or without prior acclimatization [24]). Furthermore, in most cases, for reasons of feasibility of the mountaineering expeditions or due to the limited space in HA chambers, the number of participants was low.

The aim of the current review was to systematically evaluate effects of HA exposure on cognitive performance and to explicitly categorize the applied neuropsychological tests, classifying them into their cognitive domain, and listing their results in relation to the investigated altitudes.

## 2. Materials and Methods

### 2.1. Literature Search on High-Altitude Effects on Cognitive Performance

A systematic literature review examining cognitive functions and potential hypoxic deficits at HA was performed, following the guidelines of the Preferred Reporting Items for Systematic Reviews and Meta-Analysis (PRISMA) [30].

#### 2.1.1. Search Strategy and Study Selection

A comprehensive systematic literature search was performed in the PubMed electronic database through October 2021. The search string in “all fields” was as follows: “altitude”, “normobaric hypoxia” or “hypobaric hypoxia” combined with “cognition”, “cognitive”, “neuropsychological” or “test”. First, all titles and abstracts were screened to exclude duplicate studies. Afterwards, the full text of any publication considered possibly eligible was retrieved. Additionally, articles found in the literature search were manually checked for any citations to relevant articles not listed in the electronic databases. We included only English-language studies and only studies in peer-reviewed journals obtainable in full text.

#### 2.1.2. Inclusion and Exclusion Criteria

The PICOS approach (patients, intervention, comparator, outcomes, and study design) was used to specify inclusion and exclusion criteria. The search process and the reasons for exclusion are presented in a flow diagram (Figure 1).

The inclusion criteria were studies with: (1) healthy adult subjects exposed to an altitude equal or above 2440 m (8000 ft) (either actual or through generated conditions corresponding to the intended altitude); (2) a cognitive function assessment in the context of altitude exposure using an experimental design; and (3) within-subject design with a control condition, where the baseline test condition may at most be measured up to 1500 m (4921 ft); or between-subjects design with a matched control group tested under normoxia.

Excluded were: (1) studies with a first test time point one month or longer after arrival at altitude and studies with subjects exposed to chronic hypoxia, such as miners, for the possibility of drawing conclusions from the direct effects of altitude; (2) studies that explicitly examined the effect of altitude by modulating another variable, such as sleep deprivation, exercise, or the intake of pharmaceutical agents; (3) studies performing only electrophysiological cognitive measures (e.g., event related potentials); (4) studies conducted with professional pilots or military personnel; and (5) case reports, reviews, expert opinions, comments, or letters to the editor, studies on animals, conference or abstract reports and articles not written in English.

The primary search provided the following: 1404 articles, 21 abstracts, 14 reviews, and two meta-analyses. Additionally, 30 sources were gathered from inspecting the references with the original search criteria. As a whole, 52 articles were included. On two occasions, there are two articles ([31,32,33,34]) with each referring to the same study but presenting different tests; therefore, all four articles are listed in the following. The excluded articles and their reasons for being ruled out are listed in Table A1 in Appendix B.

### 2.2. Data Extraction and Management

All publications with demographic and experimental data were screened for several factors, such as field or chamber study, hypobaric or normobaric hypoxia intervention, ascent profile, use of the STAR core parameters [35], and subject information. The following data were extracted from each included study: number, sex and mean age of participants, study protocol with experimental conditions, when given, including ascent rate, and duration of altitude or hypoxia exposure, timing and duration of neuropsychological test administration, neuropsychological test type and the related cognitive domains, and significant results indicating changes in cognitive performance.

### 2.3. Quality Assessment via STAR Data Reporting Guidelines for Clinical High-Altitude Research

The STAR data reporting guidelines were used to evaluate study quality. The aim of the STAR (Strengthening Altitude Research) initiative [35] was to create a standardized collection of core elements for research and reporting in clinical altitude medicine, and can be seen in Appendix A. The core parameters (SETTING, INDIVIDUAL FACTORS, AMS and HACE, HAPE, and TREATMENT) should help to objectivize the research reports and, thus, also support comparability.

## 3. Results

One main purpose of this review was the classification of the used neuropsychological tests and their aptitude for the assessment of cognitive function at HA.

### 3.1. Quantitative Description of the Research Methods

Of the analyzed studies, 29 studies (31 articles) were field studies and 23 were classified as laboratory studies, including two studies that investigated both conditions [36,37]. In 21 of the field studies, participants reached the altitude partially by active climbing, and in eight studies there was a passive ascent only. Of the laboratory studies, seven were performed establishing hypoxia via inhalation of a hypoxic gas mixture, eleven took place in a normobaric hypoxia chamber (NHC), and in five studies, the subjects were tested in a hypobaric hypoxia chamber (HHC). In the field investigations, the study sites were located at altitudes between 2590 m (8497 ft) [38] and 7200 m (23,622 ft) [39], with the highest point also reached at 8848 m (29,029 ft) [40]. Among the laboratory studies, the simulated altitudes under normobaric and hypobaric conditions ranged from 2440 m (8000 ft) [41] to 8848 m (29,029 ft) [42] (see also Table 1). All studies were published between 1963 and 2021.

### 3.2. Qualitative Review of the Collected Studies

STAR core parameters were assessed in all studies (Appendix A).

The use of familiarization sessions/test trials before the start of the actual cognitive assessments has been mentioned in several articles and was included in the tables. These familiarization trials are supposed to minimize learning effects or practice effects in the intraindividual comparisons. To take these learning effects over the course of several tests into account, control groups were involved in eight articles [25,28,31,36,42,47,50,67]. Learning effects were also discussed or considered in another 32 articles. In three articles, extensive familiarization trials were used to try to establish a performance plateau in advance [24,41,51]. In Lefferts et al. (2019) [51], additional training sessions occurred during the field study “*to prevent loss of task familiarity owing to the passage of time between test days*”. Another method to prevent learning effects, the use of parallel versions or alternative forms of the applied tests, was mentioned in 17 articles [25,27,31,34,40,41,43,46,49,50,53,54,56,67,72,73,74] and is marked in the following by x^1^ (meaning alternate forms used). Among them, test batteries were used in nine articles: the *Computerized Neurocognitive Test Battery CNS Vital Signs* [46,74], the *CogState Computerized Battery* [25], the *Cognition Test Battery* [43], the *PC based Multiple Attribute Task Battery* [41], the *FACTRETRIEVAL2 Test Battery* [54], the *Cambridge Neuropsychological Test Automated Battery* [31], the *ANAM-4th Edition* [72,73], and the *Defense Automated Neurobehavioral Assessment Test Battery* [34].

### 3.3. Consideration and Further Classification of the Reviewed Neuropsychological Tests

#### 3.3.1. Classification According to the Cognitive Domains Studied

As shown in Table 2, a total of 112 different neuropsychological tests were found, with the *Stroop Test* being the most frequently used with ten applications, eleven if adding the modified version. Overall, of the 173 test applications performed, 74 showed significant impairment (see also Table 3).


*Orientation*


Orientation for time and place depend on the awareness of self in relation to one’s surroundings, presuppose a consistent and reliable integration of perception and attention as well as an awareness of the ongoing narrative and, thus, memory [14]. Cognitive domains with the superordinate term orientation were tested in three studies. Here, the studies conducted in the field by Davranche et al. (2016) [24] at 4350 m (14,272 ft) and Nelson (1990) [54] at over 6500 m (21,325 ft) yielded impairments.


*Attentional capacity, processing speed and working memory*


Attention, concentration, and tracking are necessary skills for goal directed behavior. They can only be measured in the context of a cognitive activity sequence, depending on the focus. The temporary storage of information plays an essential role and a common characteristic is them having limited capacity [14]. Twelve different tests were performed to examine attentional capacity in short-term memory. Regarding the field tests, the Auditory Digit Span Test showed impairments at 4280 m (14,042 ft) [45] and at 5100 m (16,732 ft) [25], whereas other authors ([39,56]) found no impairments at higher altitudes in the field. Furthermore, the *Memory Search Task* at 4330 m (14,206 ft) [28], the *Picture Recognition Test* at 4280 m (14,042 ft) [45], and the *Verbal Free Recall Test* [55] showed reductions in performance at 4500 m (14,764 ft) as well as at 5040 m (16,535 ft). Three laboratory tests examined attention, with the *Corsi Block Forwards* and *Digit Span Test-Forward* at 4500 m (14,764 ft) [66], and the latter also at 7620 m (25,000 ft) [27] showing deteriorations.

Working memory, also referred to as mental tracking, can deal with more complex cognitive operations and allows information to be maintained in temporary storage and be manipulated [14]. This includes an executive control mechanism to focus attention and block out interference [79]. In the working memory, in 14 performed experiments, seven of them showed changes due to HA, with four tests in De Aquino’s (2012) [66] laboratory study at 4500 m (14,764 ft), namely *Corsi Block Backwards*, *Digit Span Test-Backward*, *Random Number Generation*, and *Sequence of Numbers and Letters*, showing impairments. Besides that, the *Digit Span Test-Backward* at over 5334 m (17,500 ft) [27] and the *Running Memory Continuous Performance* at 4300 m (14,107 ft) [72] also showed alterations in their first study of 2015. All five tests performed in the field showed no alterations.


*Concentration/Focused attention*


Attention in terms of concentration or focused attention is believed to be the basis for other, more complex components of cognition [14,80]. Complex attention also requires visual scanning, visuomotor coordination, motor persistence, and response speed and can be measured via symbol substitution tests [81]. Divided attention can be assessed by Trail Making Tests and it further requires scanning, visuomotor tracking, and cognitive flexibility [14]. To investigate focused attention, altitude-induced changes were examined using 20 neuropsychological tests. Limitations in performance were found in the *Code Substitution Task* at 4330 m (14,206 ft) [28], in the Continuous Performance Test at 5500 m (18,045 ft) [74], and in four investigations with the *Digit Symbol Substitution Test*, starting at 3269 m (10,725 ft) [34,49,60,74]. In contrast, improvements in the *Digit Symbol Substitution Test* were found at 5100 m (16,732 ft) [25] with no changes in two other examinations [43,46]. Using the *Frankfurt Attention Inventory-2*, subjective impaired attentional functions were reported in both field and laboratory testing [36]. In the *Paced Auditory Serial Addition Test,* the results were heterogenous, reaching from deterioration, improvement, to no change. The *Trail Making Test A* showed impaired performance in two of five studies, once in a field study at 3500 m (11,483 ft) [50] and in a laboratory study at 5334 m (17,500 ft) [27]. Two tests for response inhibition showed no changes with the *Go/No-Go* test [34,72].


*Processing speed*


Processing speed can be examined with the help of tests of response speed in regards to reaction time and accuracy data, whereas deceleration is often subject to attentional deficits [82]. Processing speed was studied in eleven experiments, approx. half of which showed no significant changes in altitude. Limitations were found in different *Reaction Time Tests*, e.g., in Subudhi et al. (2014) [34] in two investigations at 5260 m (17,257 ft). The results using the *Psychomotor Vigilance Test* showed significantly impaired performance at 3800 m (12,467 ft) [43] and at 5050 m (16,568 ft) [32]. At an altitude of 2590 m (8497 ft), Latshang et al. (2013) [38] showed no changes in the outcome of the *Psychomotor Vigilance Test*, as did Falla et al. (2021) [49] at 3269 m (10,725 ft), although deteriorations correlated with poor sleep. In Harris et al. (2009) [25], improved reaction times were found at 5100 m (16,732 ft).


*Perception*


Another topic of particular interest in HA research is visual and auditory perception. Visual perception tests require little or no physical handling of the test material. However, the complexity of brain functions makes overlap inevitable, and most tests also assess other functions such as attention, spatial orientation, or memory. Auditory perception examines, in part, skills in phoneme discrimination and speech sensation [14]. In perception, five of eight trials were unaffected by altitude [34,43,53,56]; *Reading of Briefly Displayed Letters* improved at 3450 m (11,319 ft) in both the field and the laboratory [37], whereas the *Pattern Comparison Task* showed deterioration at 4330 m (14,206 ft) [28].


*Memory*


Efficient memory can on one hand retain information, and is examined by assessing encoding using immediate retrieval trials. Second, it can retain the information even after the application of interference during a delay period. For the measurement of memorability, it is also important to understand whether underperformance is a retention or a retrieval problem [14]. For memory, three subcategories were examined in more detail in a total of eight studies. Long-term memory was not affected at altitudes up to 7100 m (23,294 ft) [54]. For verbal memory, two positive tests were found, one using *Rey’s Auditory-Verbal Learning Test* in the field at 5300 m (17,388 ft) [50], and the Verbal Memory Test in the laboratory at 5500 m (18,045 ft) [74]. Visual memory was also impaired at 5500 m (18,045 ft) [74] and on the *Match to Sample Test* at 5260 m (17,257 ft) [34].


*Verbal functions and language skills*


This heading includes tests for aphasia. Here, spontaneous speech, the ability to repeat words or sentences, syntax comprehension, the power to name things, and reading and writing are tested [14]. Two tests for verbal functions showed impairments beyond an altitude of 4340 m (14,239 ft) [56,57].


*Construction and motor performance*


This includes copying abilities, assembling and building, and, further, the assessment of motor skills and manual dexterity functions [14]. With regard to construction and motor performance, seven of twelve different tests showed impairments, although these were only examined in one study each. Two tests with multiple occurrences showed different results, such as *Finger Tapping*, with a significant deterioration in a laboratory test at 5500 m (18,045 ft) [42]. Deficits in the *Pegboard-Psychomotor Test* were shown at an altitude of 5300 m (17,388 ft) [50] and again at 5500 m (18,045 ft) [42]; however, Merz et al. (2013) [53] found unimpaired performance at 6265 m (20,555 ft) using the *Pegboard-Psychomotor Test*.


*Concept formatting and reasoning*


This category contains investigations by means of verbal procedures and visual formats such as mathematical, i.e., arithmetic reasoning problems, or, for example, by sorting and shifting [14]. For problems requiring concept formatting and reasoning, three of the six studies with six different tests were significantly altered at HA, two of them with impairment, namely in the *Category Search Task* performed at 4330 m (14,206 ft) [28] and the *Number Ordination—Rey’s Test* at 5500 m (18,045 ft) [42]. However, in the *Robinson’s Numbers Test* [57], the performance improved at HA at 4340 m (14,239 ft).


*Executive functions*


Executive functions, such as planning and decision making, are the most complex behaviors and are solid cognitive abilities with sufficient accountability necessary to respond appropriately to novel situations. At HA, the assessment of potential dangers is essential for survival. They further form the basis of many skills in the areas of cognition, emotion, and social skills. Willpower, planning and decision-making, goal-directed action, and effective performance are listed as four major components in the concept of executive functions [14]. With 22 tests and 43 experiments, executive functions were the most extensively studied cognitive domain in the collected studies. Approximately half of the experiments were associated with impairment in executive functions or increased risk behavior. In contrast, three studies found improvements or reduced risk behavior at HA. Regarding the three most common tests, the *N-Back Number Task*, *Stroop Test*, and *Trail Making Test B*, there are still mixed results. For the *N-Back Number Task*, impaired performance was found at the highest altitude tested, 5160 m (16,929 ft) [51], and in the only laboratory test at 4500 m (14,764 ft), by Williams et al. (2019) [75]. The *Stroop Test* was used ten times [23,27,46,50,52,59,62,64,66,74]. In the field examinations, the results were very heterogeneous; twice there was no disturbance [23,46], twice there was a worsening [50,59], but on the follow-up examination there was no further deviation, and once there was an improvement [52]. In the laboratory studies, four out of five tests showed impaired functions in the subjects [27,62,66,74]; the altitudes investigated ranged from 3500 m (11,483 ft) [62] to 7620 m (25,000 ft) [27]. The study by Ochi et al. (2018) [64], which investigated three altitudes from 2000 to 5000 m (6562–16,404 ft), merely found improvements in reaction time. *Trail Making Test B* revealed impairments in two of five tests, in the field above 3500 m (11,483 ft) [50] and the laboratory above 5334 m (17,500 ft) in [27]. The field study by Harris et al. (2009) [25] elicited improved results at 5100 m (16,732 ft).


*Further and mixed domains*


Lastly, the remaining domain of affective flexibility yielded mixed results in two examinations, with a significant deterioration at 4300 m (14,108 ft) [73] and no changes at 4500 m (14,764 ft) [78]. A *Mini-Mental State Examination*, screening multiple cognitive domains (concentration or working memory, language and praxis, orientation, memory, and attention span [83]), failed to detect any deficits at 4500 m (14,764 ft) [77].

**Table 2 brainsci-12-01736-t002:** Classification of neuropsychological tests into supercategories of cognitive domains supplemented by article and brief outcome.

Cognitive Domain	Neuropsychological Tests		Outcome and Altitude - *Control Condition*	First Authors
*Orientation*				
Spatiotemp-oral integration	Time Wall Estimation Task		Not affected at 3842 m (LT–HHC) - Pre-ascent BCA and post-descent CA at SL	De Bels (2019) [76]
	Time Perception Task		Impaired at 4350 (FT–P), underestimation of durations - Familiarization session, CA at SL	Davranche (2016) [24]
Confidence judgement	Feeling of Knowing		Impaired at 6500 m and 7100 (FT–A) - Pre-ascent and approx. 1 week post-descent BCA at 1200 m	Nelson (1990) [54]
*Attentional capacity, processing speed and working memory*				
Short term memory	Code (Digit Symbol) Substitution –delayed recall		Not affected at 5260 m (FT–P) - Pre-BCA (SL) 30 days prior to expedition at 130 m	Subudhi (2014) [34]
	Corsi Block Forwards		Impaired 4500 m (LT–NHC) - BCA at normoxic conditions, altitude NA	De Aquino Lemos (2012) [66]
	Digit Span Forward		Not affected at 3700 m (FT–NA) - BCA at 300 m	Zhang (2013) [60]
			Not affected at 3800 m (FT–NA) - 15-day familiarization with tests 300 ft ≙ 91 m ≙ 752 mm Hg	Phillips (1963) [57]
			Impaired at 5100 m (FT–A) - Practice test, CA at SL prior to expedition + CG, altitude NA	Harris (2009) [25]
			Not affected at 5273 m/6348 m (FT–A) - CA prior to expedition at SL and BCA on day 6 at 4000 ft ≙ 1219 m, post-expedition CA at SL between 16 and 221 days following the hypoxic exposure	Petiet (1988) [56]
			Not affected at 7200 m (FT–A) - Familiarization trial, 8 days prior to and 4 and 46 days post-expedition CA, respectively, at 1050 m	Malle (2016) [39]
			Impaired at 4500 m (LT–NHC) - BCA at normoxic conditions, altitude NA	De Aquino Lemos (2012) [66]
			Impaired at 7620 m (LT–HHC) - Pre BCA and post CA, altitude NA	Asmaro (2013) [27]
		Auditory Digit Span	Impaired at 4280 m (FT–P) - BCA “in the plain”, altitude NA	Shi (2016) [45]
		Visual Digit Span	Not affected at 4280 m (FT–P) - BCA “in the plain”, altitude NA	Shi (2016) [45]
	Memory Search Task		Impaired at 4330 m (FT–A) - Pre-ascent BCA and post-descent CA at 92 m + CG, altitude NA	Kramer (1993) [28]
	Picture Recall Test		Not affected at 4280 m (FT–P) - BCA “in the plain”, altitude NA	Shi (2016) [45]
	Picture Recognition Test		Impaired at 4280 m (FT–P) - BCA “in the plain”, altitude NA	Shi (2016) [45]
	Pocket Calculator Cognitive Motor Task		Not affected at 5400 m (FT–A) - BCA at 300 m + CG under normoxia	Bonnon (1999) [47]
	Sentence Repetition		Not affected at 3109 m/3810 m (FT–NA) - BCA at SL	Weigle (2007) [59]
	Verbal Free Recall		Impaired at 4500 m/5040 m (FT–A) - Pre-BCA, twice 15- and 40-day post-CAs	Pelamatti (2003)[55]
	Word Span Forward		Not affected at 3800 m (FT–NA) - 15-day familiarization with tests 300 ft ≙ 91 m ≙ 752 mm Hg	Phillips (1963) [57]
			Not affected at 3800 m (FT–NA) - Two groups, counterbalanced BCA at SL	Phillips (1966) [58]
Working memory	CogState: Working Memory Task Accuracy		Not affected at 5100 m (FT–A) - Practice test, CA at SL prior to expedition + CG, altitude NA	Harris (2009) [25]
	Corsi Blocks Backwards		Impaired at 4500 m (LT–NHC) - BCA at normoxic conditions, altitude NA	De Aquino Lemos (2012) [66]
	Digit Span Test Backwards		Not affected at 3700 m (FT–NA) - BCA at 300 m	Zhang (2013) [60]
			Not affected at 5100 m (FT–A) - Practice test, CA at SL prior to expedition + CG, altitude NA	Harris (2009) [25]
			Not affected at 7200 m (FT–A) - Familiarization trial, 8 days prior to and 4 and 46 days post-expedition CA, respectively, at 1050 m	Malle (2016) [39]
			Impaired at 5334 m/7620 m (LT–HHC) - Pre BCA and post CA, altitude NA	Asmaro (2013) [27]
			Impaired at 4500 m (LT–NHC) - BCA at normoxic conditions, altitude NA	De Aquino Lemos (2012) [66]
	Memory Interference Task (AB/AC paradigm)		Improved at 4000 m (LT–BHG) - Approx. 24 h apart counterbalanced blinded condition with one control CA under normoxia	Loprinzi (2019) [63]
	Operational Span Protocol with VWMC		Not affected at 3000 m (LT–NHC) - Practice trial before VWMC, BCA before entering the chamber; double blind, repeated CA under normobaric normoxia F_i_O_2_ = 20.9% near SL ≈ 113 m	Parker (2017) [68]
	Random Number Generation		Impaired at 4500 m (LT–NHC) - BCA at normoxic conditions, altitude NA	De Aquino Lemos (2012) [66]
	Running Memory Continuous Performance Test		Impaired at 4300 m (LT–NHC) - Familiarization trial and BCA under normoxia	Seo (2015) [72]
			Not affected at 4300 m (LT–NHC) - Familiarization trial and BCA under normoxia	Seo (2017) [73]
	Sequence of Numbers and Letters		Impaired at 4500 m (LT–NHC) - BCA at normoxic conditions, altitude NA	De Aquino Lemos (2012) [66]
	Sternberg’s Memory Search		Not affected at 5260 m (FT–P) - Pre-BCA 30 days prior to expedition at 130 m	Subudhi (2014) [34]
*Concentration/Focused attention*				
Attention	Attention Switching Task		Not affected at 5050 m (FT–P) - Familiarization trial, 2x BCA and post-expedition CA at 502 m + CG at 1103 m	Pun and Guadagni (2018) [31]
	Code (Letter Number) Substitution Task		Impaired at 4330 m (FT–A), insensitive to practice compared to CG - Pre-ascent BCA and post-descent CA at 92 m + CG, altitude NA	Kramer (1993) [28]
	CogState: Monitoring Task Reaction Time		Not affected at 5100 m (FT–A) - Practice test, CA at SL prior to expedition + CG, altitude NA	Harris (2009) [25]
	CogState: Monitoring Task Accuracy		Not affected at 5100 m (FT–A) - Practice test, CA at SL prior to expedition + CG, altitude NA	Harris (2009) [25]
	CogState: Working Memory Task Reaction Time		Not affected at 5100 m (FT–A) - Practice test, CA at SL prior to expedition + CG, altitude NA	Harris (2009) [25]
	Colorado Perceptual Speed Test		Not affected at 5300 m (FT–A) - Pre-ascent BCA and post-descent CA at 1400 m	Karinen (2017) [40]
	Continuous Performance Test		Not affected at 5500 m (LT–BHG) - Pre BCA under normoxia, post CA	Altbäcker (2019)[61]
			Impaired at 5500 m (LT–NHC) - BCA under normoxia, matched-pairs study with a single-blind, randomized design, sham group F_i_O_2_ = 21% ≙ SL	Turner (2015) [74]
	Digit Symbol Substitution Test		Impaired at 3269 m—session 3 (FT–A) - Familiarization session, BCA at 1258 m	Falla (2021) [49]
			Impaired at 3700 m (FT–NA) - BCA at 300 m	Zhang (2013) [60]
			Not affected at 3800 m (FT–P) - Practice sessions, group split with pre-ascent BCA or post-descent CA at 340 m ASL	Frost (2021) [43]
			Not affected at 4554 m (FT–A) - Pre-study: one night at 2864 m, familiarization session, pre BCA and post CA at 1115 m	Bjursten (2010) [46]
			Improved at 5100 m (FT–A) - Practice test, CA at SL prior to expedition + CG, altitude NA	Harris (2009) [25]
			Impaired at 5260 m (FT–P) - Pre-BCA 30 days prior to expedition at 130 m	Subudhi (2014) [34]
			Impaired at 5500 m (LT–NHC) - BCA under normoxia, matched-pairs study with a single-blind, randomized design, sham group F_i_O_2_ = 21% ≙ SL	Turner (2015) [74]
	Divided Attention Steering Simulator		Not affected at 2590 m (FT–P) - Pre-ascent BCA and post-descent CA at 490 m	Latshang (2013) [38]
	Eriksen Flanker		Not affected at 5160 m (FT–A) - Extensive familiarization process prior to the trek, and practice tests throughout the trek; CA prior to expedition at 116 m ASL	Lefferts (2019) [51]
			Not affected at 4500 m (LT–NHC) - Familiarization session; CA under SL conditions; BCA prior to exposure under supply of normoxic air	Williams (2019) [75]
	Frankfurt Attention Inventory-2		Impaired at 5339 m (FT–A) - CA prior to expedition at 154 m and post-descent CA at 812 m + CG under normoxia + CG with 7 days of physical exercise under normoxia	Limmer (2018) [36]
			Impaired at 5800 m (LT–NHC) - Pre- and post-CA at 53 m normoxic conditions + CG under normoxia + CG with 7 days of physical exercise under normoxia	
	Letter Cancellation Test		Impaired at 3500 m/5300 m (FT–A) - BCA at 75 m prior to expedition + CG at/or near SL, altitude NA	Griva (2017) [50]
	Paced Auditory Serial Addition Test		Impaired at 4280 m (FT–P) - BCA “in the plain”, altitude NA	Shi (2016) [45]
			Improved at 5273 m/6348 m (FT–A) - CA prior to expedition at SL and BCA on day 6 at 4000 ft ≙ 1219 m, post-expedition CA at SL between 16 and 221 days following the hypoxic exposure	Petiet (1988) [56]
			Not affected at 7200 m (FT–A) - Familiarization trial, 8 days prior to and 4 and 46 days post-expedition CA, respectively, at 1050 m	Malle (2016) [39]
	Paced Visual Serial Addition Test		Impaired at 4280 m (FT–P) - BCA “in the plain”, altitude NA	Shi (2016) [45]
	Rapid Visual Processing Test		Not affected at 5050 m (FT–P) - Familiarization trial, 2x BCA and post-expedition CA at 502 m + CG at 1103 m	Pun and Guadagni (2018) [31]
	Ruff 2 and 7 Selective Attention Test		Not affected at 6265 m (FT–A) - Familiarization trial, prior to and 3 months post-expedition CA at 440 m	Merz (2013) [53]
	Selective Auditory Attention Task		Not affected at 5273 m/6348 m (FT–A) - CA prior to expedition at SL and BCA on day 6 at 4000 ft ≙ 1219 m, post-expedition CA at SL between 16 and 221 days following the hypoxic exposure	Petiet (1988) [56]
	Simple Reaction Time Test Span dSRT		Impaired at 5260 m (FT–P) - 30 days prior to and 3 months post-expedition CA at SL	Roach (2014) [33]
	Symbol Digit Modalities test		Impaired at 3500 m/5300 m (FT–A) - BCA at 75 m prior to expedition + CG at/or near SL, altitude NA	Griva (2017) [50]
	Trail Making Test A		Not affected at 2590 m (FT–P) - Pre-ascent BCA and post-descent CA at 490 m	Latshang (2013) [38]
			Impaired at 3500 m/5300 m (FT–A) - BCA at 75 m prior to expedition + CG at/or near SL, altitude NA	Griva (2017) [50]
			Not affected at 5050 m (FT–P) - Familiarization trial, 2x BCA and post-expedition CA at 502 m	Pun and Hartmann (2018) [32]
			Not affected at 5500 m (FT–A) - Familiarization trial, pre-ascent BCA and post-descent CA at 1400 m, retesting after return	Issa (2016) [23]
			Impaired at 5334 m/7620 m (LT–HHC) - Pre BCA and post CA, altitude NA	Asmaro (2013) [27]
Response Inhibition	Go/No-Go Test		Not affected at 4300 m (LT–NHC) - Familiarization trial and BCA under normoxia	Seo (2015) [72]
			Not affected at 5260 m (FT–P) - Pre-BCA 30 days prior to expedition at 130 m	Subudhi (2014) [34]
*Processing speed*				
Reaction time	CogState: Simple Reaction Time		Improved at 5100 m (FT–A) - Practice test, CA at SL prior to expedition + CG, altitude NA	Harris (2009) [25]
	Deary–Liewald Reaction Time Task		Not affected at 4500 m (LT–NHC) - Familiarization session; CA under SL conditions; BCA prior to exposure under supply of normoxic air	Williams (2019) [75]
	Perceptual Vigilance task		Not affected at 3842 m (LT–HHC) - Pre-ascent BCA and post-descent CA at SL	De Bels (2019) [76]
	Procedural Reaction Time		Impaired at 5260 m (FT–P) - Pre-BCA 30 days prior to expedition at 130 m	Subudhi (2014) [34]
	Psychomotor Vigilance Test		Not affected at 2590 m (FT–P) - Pre-ascent BCA and post-descent CA at 490 m	Latshang (2013) [38]
			Not affected at 3269 m (FT–A) but higher impairment for poor sleepers - Familiarization session, BCA at 1258 m	Falla (2021) [49]
			Impaired at 3800 m (FT–P) - Practice sessions, group split with pre-ascent BCA or post-descent CA at 340 m ASL	Frost (2021) [43]
			Impaired at 5050 m (FT–P) - Familiarization trial, 2x BCA and post-expedition CA at 502 m	Pun and Hartmann (2018) [32]
	Reaction Time Task		Not affected at 5050 m (FT–P) - Familiarization trial, 2x BCA and post-expedition CA at 502 m + CG at 1103 m	Pun and Guadagni (2018) [31]
	Simple Reaction Time Test		Not affected at 3700 m (FT–NA) - BCA at 300 m	Zhang (2013) [60]
			Impaired at 5260 m (FT–P) - Pre-BCA 30 days prior to expedition at 130 m	Subudhi (2014) [34]
*Perception*				
Visual perception	Line Bisection Test		Not affected at 6265 m (FT–A) - Familiarization trial, prior to and 3 months post-expedition CA at 440 m	Merz (2013) [53]
	Reading of Briefly Displayed Letters		Improved at 3450 m (FT–P) - Familiarization trial, BCA at 540 m	Schlaepfer (1992) [37]
			Improved at 3450 m (LT–BHG) - Familiarization trial, BCA at 540 m	
Visuospatial analytic ability	Abstract Matching		Not affected at 3800 m (FT–P) - Practice sessions, group split with pre-ascent BCA or post-descent CA at 340 m ASL	Frost (2021) [43]
	Line Orientation Task		Not affected at 3800 m (FT–P) - Practice sessions, group split with pre-ascent BCA or post-descent CA at 340 m ASL	Frost (2021) [43]
			Not affected at 5273 m/6348 m (FT–A) - CA prior to expedition at SL and BCA on day 6 at 4000 ft ≙ 1219 m, post-expedition CA at SL between 16 and 221 days following the hypoxic exposure	Petiet (1988) [56]
	Pattern Comparison Task		Impaired at 4330 m (FT–A) - Pre-ascent BCA and post-descent CA at 92 m + CG, altitude NA	Kramer (1993) [28]
	Spatial Discrimination		Not affected at 5260 m (FT–P) - Pre-BCA 30 days prior to expedition at 130 m	Subudhi (2014) [34]
*Memory*				
Long term memory	FACTRETRIEVAL2 Test Battery		Not affected at 5400 m/6500 m/7100 m (FT–A) - Pre-ascent and approx. 1 week post-descent BCA at 1200 m	Nelson (1990) [54]
Verbal memory	Rey’s Auditory–Verbal Learning Test		Not affected at 5100 m (FT–A) - Practice test, CA at SL prior to expedition + CG, altitude NA	Harris (2009) [25]
			Impaired at 5300 m (FT–A) - BCA at 75 m prior to expedition + CG at/or near SL, altitude NA	Griva (2017) [50]
	Selective Reminding Test		Not affected at 5273 m/6348 m (FT–A) - CA prior to expedition at SL and BCA on day 6 at 4000 ft ≙ 1219 m, post-expedition CA at SL on between 16 to 221 days following the hypoxic exposure	Petiet (1988) [56]
	Verbal memory test		Not affected at 4554 m (FT–A) - Pre-study: one night at 2864 m, familiarization session, pre BCA and post CA at 1115 m	Bjursten (2010) [46]
			Impaired at 5500 m (LT–NHC) - BCA under normoxia, matched-pairs study with a single-blind, randomized design, sham group F_i_O_2_ = 21% ≙ SL	Turner (2015) [74]
Visual memory	CogState: Learning Task Accuracy		Not affected at 5100 m (FT–A) - Practice test, CA at SL prior to expedition + CG, altitude NA	Harris (2009) [25]
	Match to sample		Impaired at 5260 m (FT–P) - Pre-BCA 30 days prior to expedition at 130 m	Subudhi (2014) [34]
	Visual Memory Test		Not affected at 4554 m (FT–A) - Pre-study: one night at 2864 m, familiarization session, pre BCA and post CA at 1115 m	Bjursten (2010) [46]
			Impaired at 5500 m (LT–NHC) - BCA under normoxia, matched-pairs study with a single-blind, randomized design, sham group F_i_O_2_ = 21% ≙ SL	Turner (2015) [74]
	Visual Object Learning Task		Not affected at 3800 m (FT–P) - Practice sessions, group split with pre-ascent BCA or post-descent CA at 340 m ASL	Frost (2021) [43]
*Verbal functions and language skills*				
Speech production and syntax comprehension	Boston Naming Test		Impaired at 5273 m/6348 m (FT–A) - CA prior to expedition at SL and BCA on day 6 at 4000 ft ≙ 1219 m, post-expedition CA at SL between 16 and 221 days following the hypoxic exposure	Petiet (1988) [56]
	Robinson’s Rhymes (and Numbers) tests		Impaired at 4340 m (FT–NA) - Two groups, counterbalanced BCA at SL	Phillips (1963) [57]
*Construction and motor performance*				
Copying	Benton Visual Retention Test		Not affected at 3700 m (FT–NA) - BCA at 300 m	Zhang (2013) [60]
Assembling and building	Block Design		Impaired at 3500 m/5300 m (FT–A) - BCA at 75 m prior to expedition + CG at/or near SL, altitude NA	Griva (2017) [50]
Psychomotor ability	Computer-based Psychomotor Speed Task		Impaired at 3000 m (LT–NHC) - Familiarization session, CA under normoxia F_i_O_2_ = 20.9 % ≙ 0 m ASL in randomized order	Pighin (2012) [70]
	Finger Tapping		Not affected at 4330 m (FT–A) - Pre-ascent BCA and post-descent CA at 92 m + CG, altitude NA	Kramer (1993) [28]
			Not affected at 4554 m (FT–A) - Pre-study: one night at 2864 m, familiarization session, pre BCA and post CA at 1115 m	Bjursten (2010) [46]
			Not affected at 5273 m/6348 m (FT–A) - CA prior to expedition at SL and BCA on day 6 at 4000 ft ≙ 1219 m, post-expedition CA at SL between 16 and 221 days following the hypoxic exposure	Petiet (1988) [56]
			Impaired at 5500 m (LT–NHC) - BCA under normoxia, matched-pairs study with a single-blind, randomized design, sham group F_i_O_2_ = 21% ≙ SL	Turner (2015) [74]
	Motor Praxis Task		Not affected at 3800 m (FT–P) - Practice sessions, group split with pre-ascent BCA or post-descent CA at 340 m ASL	Frost (2021) [43]
	Pegboard Psychomotor Test		Impaired at 5300 m (FT–A) - BCA at 75 m prior to expedition + CG at/or near SL, altitude NA	Griva (2017) [50]
			Not affected at 6265 m (FT–A) - Familiarization trial, prior to and 3 months post-expedition CA at 440 m	Merz (2013) [53]
			Impaired at 5500 m (LT–HHC) - 3x Pre BCA and 1x Post CA at SL + CG at SL	Abraini (1998) [42]
	Pursuit Aiming Test		Impaired at 3700 m (FT–NA) - BCA at 300 m	Zhang (2013) [60]
	Santa Ana Manual Dexterity Test		Impaired at 3700 m (FT–NA) - BCA at 300 m	Zhang (2013) [60]
	Surgical Skills		Impaired at 3000 m (LT–NHC) - BCA before entering the chamber; double blind, repeated CA under normobaric normoxia FiO2 = 20.9% near SL ≈ 113 m	Parker (2017) [68]
	Visual Motor Reaction Time		Impaired at 3109 m/3810 m (FT–NA) - BCA at SL	Weigle (2007) [59]
Ocular motor performance	King–Devick Test		Impaired at 7101 m (LT–BHG) - Familiarization tests, BCA and 3 min post CA at normoxia	Stepanek (2013) [65]
	Saccadic Eye Movement		Not affected at 6265 m (FT–A) - Familiarization trial, prior to and 3 month post-expedition CA at 440 m	Merz (2013) [53]
*Concept formatting and Reasoning*				
Concept formation	Category Search Task		Impaired at 4330 m (FT–A) - Pre-ascent BCA and post-descent CA at 92 m + CG, altitude NA	Kramer (1993) [28]
Mental Efficacy, reasoning	Gorham’s Proverbs		Not affected at 5273 m/6348 m (FT–A) - CA prior to expedition at SL and BCA on day 6 at 4000 ft ≙ 1219 m, post-expedition CA at SL between 16 and 221 days following the hypoxic exposure	Petiet (1988) [56]
	Number Ordination—Rey’s test		Impaired at 5500 m (LT–HHC) - 3x Pre BCA and 1x Post CA at SL + CG at SL	Abraini (1998) [42]
	Robinson’s (Rhymes and) Numbers tests		Improved at 4340 m (FT–NA) - Two groups, counterbalanced BCA at SL	Phillips (1963) [57]
	Verbal Reasoning Test		Not affected at 3109 m/3810 m (FT–NA) - BCA at SL	Weigle (2007) [59]
Arithmetic Reasoning Problems	Modified Math Processing Task		Not affected at 3842 m (LT–HHC) - Pre-ascent BCA and post-descent CA at SL	De Bels (2019) [76]
*Executive functions*				
Executive function	Attention Shifting Test		Not affected at 4554 m (FT–A) - Pre-study: one night at 2864 m, familiarization session, pre BCA and post CA at 1115 m	Bjursten (2010) [46]
			Impaired at 5500 m (LT–NHC) - BCA under normoxia, matched-pairs study with a single-blind, randomized design, sham group F_i_O_2_ = 21 % ≙ SL	Turner (2015) [74]
	Category Fluency Tasks		Not affected at 3000 and 4500 m (LT–HHC) - Control profile at 450 m with 200 m “pseudo-ascent” to mask for altitude	Pavlicek (2005) [78]
	Controlled Oral Word Association		Not affected at 5100 m (FT–A) - Practice test, CA at SL prior to expedition + CG, altitude NA	Harris (2009) [25]
			Impaired at 3500 m/5300 m (FT–A) - BCA at 75 m prior to expedition + CG at/or near SL, altitude NA	Griva (2017) [50]
	Choice Reaction Test (Schuhfried)		Impaired at 3500 m and 5500 m (LT–NHC) - BCA at 450 m ≙ F_i_O_2_ = 20.93%	Pramsohler (2017) [71]
	CogState: Choice Reaction Time		Not affected at 5100 m (FT–A) - Practice test, CA at SL prior to expedition + CG, altitude NA	Harris (2009) [25]
	Four-Choice Reaction Time		Impaired at 4000 m/5565 m (FT–A) - 4 x CA at 76–86 m ASL prior to expedition	Dykiert (2010) [48]
	MATB–Multiple Attribute Task Battery		Impaired at 2440 m (LT–BHG) - BCA under normoxia with two-week training to establish a personal performance plateau, post-exposure CA under 100% O_2_	Kourtidou-Papadeli (2008) [41]
	N-Back Number Task		Not affected at 2590 m (FT–P) - Pre-ascent BCA and post-descent CA at 490 m	Latshang (2013) [38]
			Not affected at 3800 m (FT–P) - Practice sessions, group split with pre-ascent BCA or post-descent CA at 340 m ASL	Frost (2021) [43]
			Impaired at 5160 m (FT–A), improved reaction time - Extensive familiarization process prior to the trek, and practice tests throughout the trek; CA prior to expedition at 116 m ASL	Lefferts (2019) [51]
			Impaired at 4500 m (LT–NHC) - Familiarization session; CA under SL conditions; BCA prior to exposure under supply of normoxic air	Williams (2019) [75]
	Number Comparison Test		Not affected at 5300 m (FT–A) - Pre-ascent BCA and post-descent CA at 1400 m	Karinen (2017) [40]
	One Touch Stockings of Cambridge Task		Not affected at 5050 m (FT–P) - Familiarization trial, 2x BCA and post-expedition CA at 502 m + CG at 1103 m	Pun and Guadagni (2018) [31]
	Pro-Point and Anti-Point Tasks		Not affected at 4330 m (FT–P) - Familiarization test, BCA at 344 m prior to expedition	Gibbons (2020) [44]
	Rapid Cognitive Assessment Tool		Not affected at 5500 m (FT–A) - Familiarization trial, pre-ascent BCA and post-descent CA at 1400 m, retesting after return	Issa (2016) [23]
	Ruff Figural Fluency Test		Not affected at 6265 m (FT–A) - Familiarization trial, prior to and 3 months post-expedition CA at 440 m	Merz (2013) [53]
	Simon Task		Impaired at 4350 (FT–P) - Familiarization session, CA at SL	Davranche (2016) [24]
	Stroop Test		Impaired at 3109 m, not affected at 3810 m (FT–NA) - BCA at SL	Weigle (2007) [59]
			Impaired at 3500 m/5300 m (FT–A) - BCA at 75 m prior to expedition + CG at/or near SL, altitude NA	Griva (2017) [50]
			Improved at 4240 m (FT–A) - CA prior to expedition at 116 m ASL	Lefferts (2020) [52]
			Not affected at 4554 m (FT–A) - Pre-study: one night at 2864 m, familiarization session, pre BCA and post CA at 1115 m	Bjursten (2010) [46]
			Not affected at 5500 m (FT–A) - Familiarization trial, pre-ascent BCA and post-descent CA at 1400 m, retesting after return	Issa (2016) [23]
			Impaired at 3500 m (LT–BHG) - Familiarization session, CA under normoxia, altitude NA	Chroboczek (2021) [62]
			Impaired 4500 m (LT–NHC) - BCA at normoxic conditions, altitude NA	De Aquino Lemos (2012) [66]
			Not affected at 2000 m/3500 m/5000 m (LT–BHG), improved reaction time - Two practice sessions, CA at SL	Ochi (2018) [64]
			Impaired at 5500 m (LT–NHC) - BCA under normoxia, matched-pairs study with a single-blind, randomized design, sham group F_i_O_2_ = 21% ≙ SL	Turner (2015) [74]
			Impaired at 7620 m (LT–HHC) - Pre BCA and post CA, altitude NA	Asmaro (2013) [27]
		Number-Size Stroop Variant Task	Not affected at 5500 m (LT–BHG) - Pre BCA under normoxia, post CA	Altbäcker (2019) [61]
	Trail Making Test B		Impaired at 3500 m and 5300 m (FT–A) - BCA at 75 m prior to expedition + CG at/or near SL, altitude NA	Griva (2017) [50]
			Not affected at 5050 m (FT–P) - Familiarization trial, 2x BCA and post-expedition CA at 502 m	Pun and Hartmann (2018) [32]
			Improved at 5100 m (FT–A) - Practice test, CA at SL prior to expedition + CG, altitude NA	Harris (2009) [25]
			Not affected at 5500 m (FT–A) - Familiarization trial, pre-ascent BCA and post-descent CA at 1400 m, retesting after return	Issa (2016) [23]
			Impaired at 5334 m/7620 m (LT–HHC) - Pre BCA and post CA, altitude NA	Asmaro (2013) [27]
	Verbal Letter Fluency		Not affected at 3000/4500 m (LT–HHC) - Control profile at 450 m with 200 m “pseudo-ascent” to mask for altitude	Pavlicek (2005) [78]
	Visual Choice Reaction Time		Impaired at 4330 m (FT–A) - Pre-ascent BCA and post-descent CA at 92 m + CG, altitude NA	Kramer (1993) [28]
			Not affected at 6500 m (LT–HHC) - 3x Pre BCA and 1x Post CA at SL + CG at SL	Abraini (1998) [42]
Planning and decision making	Balloon Analogue Risk Taking task		Risk-taking not affected at 3269 m—session 4 (FT–A) with faster reaction times and higher earnings - Familiarization session, BCA at 1258 m	Falla (2021) [49]
			Risk-taking not affected at 3800 m (FT–P) with faster reaction times - Practice sessions, group split with pre-ascent BCA or post-descent CA at 340 m ASL	Frost (2021) [43]
			Impaired, higher risk taking at 3000 m (LT–NHC) - Familiarization session, CA under normoxia F_i_O_2_ = 20.9% ≙ 0 m ASL in randomized order	Pighin (2019) [69]
	Computer-based Risk-Taking Task		Increased risk-taking for choices involving losses at 3000 m (LT–NHC) - Familiarization session, CA under normoxia F_i_O_2_ = 20.9% ≙ 0 m ASL in randomized order	Pighin (2012) [70]
	Game of Dice Task		Increased risk behavior at 4500 m (LT–NHC) with reduced risk behavior in preacclimatized subjects compared to unacclimatized subjects - Familiarization session, CG with sham pre-acclimatization 7 × 1 h at FiO_2_ = 20.9% ≙ 600 m ASL	Niedermeier (2017) [67]
*Further and mixed domains*				
Affective flexibility	Lateralized Tachistoscopic Lexical Decision Task		Not affected at 3000/4500 m (LT–HHC) - Control profile at 450 m with 200 m “pseudo-ascent” to mask for altitude	Pavlicek (2005) [78]
	Total Mood Disturbance		Impaired at 4300 m (LT–NHC) - Familiarization trial and BCA under normoxia	Seo (2017) [73]
Unspecified individual test outcome	Mini-Mental State Examination: concentration or working memory, language and praxis, orientation, memory, attention span, and other cognitive factors		Not affected at 4500 m (LT–HHC) - No control condition	Nakano (2015) [77]

Legend. A: Active mode of ascent; ASL: Above sea level; BCA: Baseline cognitive assessment; BHG: Breathing of hypoxic gas mixture; CA: Cognitive assessment; CG: Control group; FT: Field test; HHC: Hypobaric hypoxia chamber; LT: Laboratory test; NA: Mode of ascent/information not available; NHC: Normobaric hypoxia chamber; P: Passive mode of ascent; SL: Sea level. Note. The neuropsychological tests are grouped according to the cognitive domains studied, with a brief result and the study conditions. Where the same tests have been used, the field studies are mentioned before the laboratory studies and are ordered by increasing altitudes. The mode of ascent is given in parentheses. Neuropsychological tests were merged if they were likely identical with a slight change in name (e.g., the *Digit Symbol Substitution Test (DSST)* was named *Symbol Digit Substitution Test* in one study).

**Table 3 brainsci-12-01736-t003:** Frequency of neuropsychological test results depending on research condition.

**Research Condition**	**Neuropsychological Test Application** **N**			
	**Not affected**	**Impairment**	**Improvement**	**Total**
FT	74	41	7	122
A	41	22	5	68
NA	9	6	1	16
P	24	13	1	38
*M_altitude_ FT*	4870 m (15,978 ft)	4364 m (4364 ft)	4811 m (15,784 ft)	4696 m (15,407 ft)
**Research condition**	**Not affected**	**Impairment**	**Improvement**	**Total**
LT	16	33	2	51
BHG	3	3	2	8
HHC	8	7		15
NHC	5	23		28
*M_altitude_ LT*	4539 m (14,892 ft)	4748 m (15,577 ft)	3725 m (12,221 ft)	4642 m (15,230 ft)

Legend. A: Active mode of ascent; BHG: Breathing of hypoxic gas mixture; FT: Field test; HHC: Hypobaric hypoxia chamber; LT: Laboratory test; NA: Mode of ascent not specified; NHC: Normobaric hypoxia chamber; P: Passive mode of ascent. Looking back at Table 1, studies conducted in the field were slightly superior in number with 29 expeditions and 31 articles compared to those 23 conducted in the laboratory. Regarding test applications in Table 3, more than twice as many took place in the field than in the laboratory.

#### 3.3.2. Classification According to the Numerical Frequency of Results

Table 3 below provides an overview of the frequency of results depending on the research conditions, with test results divided into no alteration, impairment, and improvement. In the literature, the improvements listed in the tables are most likely attributed to learning effects from the repeated measurements and are not further considered here for simplification reasons. The numbers from Table 3 suggest that the measured impairments might have something to do with the rate of acclimatization. Mean values were calculated in relation to the altitudes showing impairment in field (1) versus laboratory studies (2). Regarding the results showing impairment on several levels of altitude, the lowest significant level was chosen. Due to different variances, a *t*-Test assuming unequal variances was calculated, and the mean altitudes did not differ (3).
*M_altitude_* (impairment field tests) = 4364 m (4364 ft), (*range* 3109–6500 m [10,200–21,325 ft])(1)
*M_altitude_* (impairment chamber tests) = 4748 m (15,577 ft), (*range* 2440–7620 m [8000–25,000 ft])(2)
*S^2^_field_* = 652,322; *S*^2^*_chamber_* = 1336,602; *p* = 0.0056(3)

Of the 173 tests applied (sum of field and laboratory tests), 70.5% were conducted in the field. Of the field tests, slightly more than half of the total 122 test applications were performed under active ascent. Of all tests performed in the field, 60.7% showed no impairment, 5.7% showed improvement, and 33.6% found significant differences at HA. In active ascent, 60.3% showed no alteration, and 32.4% showed deterioration. In passive ascent, 63.2 % of the trials had no alterations, and 34.2% had deteriorations. Of the 51 laboratory tests, 31.4% showed no change, in 3.9% there was an improvement, and 64.7% showed deterioration in percentage terms. Numerically, in hypoxic gas mixture testing and hypobaric chamber studies, no impairment occurred about as often as impairment. Of the studies with normobaric hypoxia, 82.1 % showed hypoxia-related deterioration.

### 3.4. Overview of the Results

Figure 2A provides an overview of the test results in field studies, and Figure 2B the test results in laboratory studies, divided into impairment, no affection, and improvement, and plotted against altitude. Regarding the results showing impairment on several levels of altitude, the lowest significant level was chosen. For results without affections at altitude, the highest level of altitude was selected.

## 4. Discussion

The aim of the review was not only to report an overview of the study results and summarize the evidence for and against cognitive impairment at moderate, high and extreme altitudes, but also to provide a closer look at the neuropsychological tests used. For this purpose, they were grouped under the respective cognitive domains and the results of each neuropsychological test were listed in detail.

### 4.1. Major Findings

The major findings of the current analysis were that 112 different tests have been used. With 74 of 173 test applications per subject, less than half of the tests resulted in impairment. A novel approach in this review was to assign each neuropsychological test to its cognitive domain. Since some brain regions appear to be more dependent on oxygen, it would be interesting to see whether this also manifests itself in differences in the individual cognitive domains. Attentional capacity, concentration, and executive functions were the most frequently studied cognitive domains. However, it is not possible to conclude that they were more sensitive than others, because we found impairments in all cognitive domains.

In the most extensively studied cognitive domain “*Attentional capacity, processing speed and working memory*”, 5 out of 21 field studies yielded impaired results. However, it is noticeable that out of the thirteen laboratory tests, nine found deteriorations above the simulated altitude of 4300 m (14,108 ft). It seems possible that some cognitive domains are more likely to be distracted by sudden hypoxic conditions than others, as might be the case in the laboratory studies.

#### 4.1.1. Discussion of Threshold Altitude

An obvious conclusion would be that there are preconditions, such as a threshold altitude, under which significant differences collected with a particular test can be reproduced. A good example that this is partially true can be found in the *Psychomotor Vigilance Test*, which was used in four field studies. In an altitude-dependent order of the results, two, namely the studies of Frost et al. (2021) [43] at 3800 m (12,467 ft) and, further, by Pun and Hartmann (2018) [32] at 5050 m (16,568 ft), show impairments. The investigations of reaction time using the *Psychomotor Vigilance Test* at 2590 m (8497 ft) [38] and at 3269 m (10,725 ft) [43] show no impairments. Therefore, a logical conclusion would be to assume that the subjects’ ability to respond at sea level are limited with an increase in the altitude of the study. Regarding the *Psychomotor Vigilance Test*, in particular, this would be between an altitude of 3269 m (10,725 ft) and 3800 m (12,467 ft). The picture is similar for the *N-Back Number Task*, which was used in four tests and twice showed an impairment at HA. The field test of Frost et al. (2021) [43] at 3800 m (12,467 ft) was without changes, whereas the laboratory test of Williams et al. (2019) [75] at 4500 m (14,764 ft) shows significant differences.

That this “threshold altitude” cannot be generalized to other tests appears evident, with reference to the most extensively studied cognitive domain of executive functions. In this domain, a large number of different tests was used, some of which showed no impairment up to very high altitudes, such as the *Ruff Figural Fluency Test* of Merz et al. (2013) [53] at 6265 m (20,555 ft) in the field.

The *Stroop Task* is the most represented with ten applications and an overview of the results of the studies concerned raises new questions. Namely, of the five field tests, the study conducted at the lowest point, 3109 m (10,200 ft), by Weigle et al. (2007) [59], deviates from the norm, whereas the study conducted at the highest point, 5500 m (18,045 ft), by Issa et al. (2016) [23] is found to be unaffected.

#### 4.1.2. Effects of the Ascent Mode

The mode of ascent could influence the results of neurocognitive tests at HA. Roughly broken down, the passive mode of ascent using cars or mountain cable cars shows higher ascent rates than for an active ascent on foot. As described before, a slower ascent contributes to better acclimatization [10,84]. With reference to Davrache et al. (2016) [24], better acclimatization would result in fewer negative effects caused by HA. In laboratory tests, in most cases, hypoxic conditions were established much faster than they would be via active ascent in field studies. Looking again at the *Stroop Test*, it is noticeable that impairments were found in four out of five laboratory tests, starting at 3500 m (11,483 ft) [62], up to 7620 m (25,000 ft) [74].

In addition, there seem to be other issues such as test–retest reliability, which can be seen unprecedentedly in the two studies by Seo et al. (2015 and 2017). For example, the first study in 2015 [72] showed significant changes in the *Running Memory Continuous Performance Test*, but the re-run in 2017 [73] could not replicate this result. If mathematical significance calculations can exclude accidental events, there must be other influencing mediators that have not yet been conclusively clarified. The obvious and already extensively researched ones would be, for example, physical strain, cold, or the quality of sleep.

#### 4.1.3. Differences of Field and Chamber Test Applications and Consequent Assumptions

Far more than twice as many test applications took place in the field and there is a difference in the number of significant results. Proportionally, the tests that found impairments in the field at HA amounted to about one third of all tests conducted in the field and slightly less than two thirds showed no changes. In contrast, for laboratory tests, the ratio of tests without altitude-induced changes and those with impairments seemed to be almost reversed. Since there was no difference between the calculated mean altitude of impairments in field versus laboratory test settings, it is yet unclear what exactly is the reason for cognitive performance being proportionally more often impaired in laboratory studies than in field studies. The more rapid onset of compromising circumstances and insufficient acclimatization could be held accountable. As mentioned previously, Martin et al. [10] had concluded that the duration of the altitude exposure and altitude level had the greatest impact on cognition, and that altitude acclimatization appeared to have a positive effect on cognitive performance. In field studies, the ascent rate is decisively lower compared to that of passive field tests. In addition, the overall duration is longer than in most laboratory studies, which means that acclimatization can take place over a longer period. In conclusion, this review also suggests that sufficient acclimatization has a beneficial effect on cognitive functions at HA.

Alongside this, however, it must also be mentioned that in different experimental conditions, different stressors are in the foreground as well. Besides hypoxia, supplemental stressors in the field may include physical exhaustion from the climb, cold, sleep disturbance, and psychological factors, such as stress and anxiety, over the climb. Participants in passive ascents are exposed to hypoxic conditions more rapidly, which is why the problems of inadequate acclimatization are more likely to occur, as measured by the Lake Louise Scale, which is designed for the clinical diagnosis of AMS and assessment of its severity [85]. In laboratory studies, it is probably also the latter. Additionally, in chamber studies, difficulties were caused by unfamiliar situations, such as the confined space that allows only limited freedom of movement and isolation from the other group members, not to neglect psychological factors, such as claustrophobia. The extent to which these specific factors affect the study results is unclear at this time and needs to be considered.

### 4.2. Limitations

Throughout this review, an attempt has been made to give space to the complexity of the studies analyzed and to list the research modalities and results as completely as possible. On the other hand, the aim was to approach the object of investigation, the effects of hypoxia on cognitive abilities, as straightforwardly and comprehensibly as possible. These intentions are, in view of modern research with consideration of numerous cofactors in the statistical models, contrary to a simple approach. It is important to mention the risk of potential bias in this work. For one thing, this may be the case due to the so-called “publication bias”, i.e., the potentially selective reporting of complete studies and non-publication of study results without significant findings. However, “outcome reporting bias” within individual studies also has to be discussed as a reason for sources of error.

Despite the broad sample of studies, it must also be mentioned that a large number of studies, some of them highly qualitative, were excluded because they were conducted with military personnel or aircrew. The decision against including studies with professional pilots or military personnel is based on the fact that, on the one hand, highly specialized individuals, e.g., air combat pilots, seem difficult to compare with mountaineers. Second, the studies were often computer-based multi-tasking assessments (such as those used in the study of Kourdidou-Papadeli et al. [41]) that examined occupation-specific scenarios and made it difficult to draw direct conclusions about underlying cognitive functions due to the multifactorial and comprehensive nature of the assignments.

The bundling of the test procedures into cognitive domains was conducted by drawing artificial dividing lines that are not entirely objectively reproducible. As already noted in the methods section, this artificial simplification is a source of error. One possibility to at least contain it or to standardize it across the studies would also be the creation of a standardized framework of tests. One difficulty and possible bias in classifying the tests was that some of the same neuropsychological tests were used, but their names differed somewhat. In places where this was clearly evident, summaries were made under the same test name. However, this was probably not successful everywhere, which is one more reason to use standardized test batteries.

## 5. Conclusions

This review explicitly reports on applied neuropsychological tests, classifies them into cognitive domains, and correlates their results to the investigated altitudes and the intervention methods. Altitude and duration of altitude exposure seem to have the greatest impact on cognition, and sufficient acclimatization has beneficial effects on cognitive functions at HA.

Future studies should try to find a common consensus and complement each other. One possibility would be the creation of a standardized framework of tests, for instance an open access test battery. If used frequently, a standardized test battery could make the different experiments more comparable and help to identify fundamental influencing factors. To date, there have been studies with computerized test batteries; however, these were provided by commercial service companies and were therefore only used in occasional studies.

The “STAR” data reporting guidelines provide investigators with a suggested pathway of quality aspects to examine. Inter-study comparisons can be made more easily. The STAR guidelines for data reporting have proved useful in assessing study quality, and their utilization in future studies may contribute to a further standardization of methods.

## Figures and Tables

**Figure 1 brainsci-12-01736-f001:**
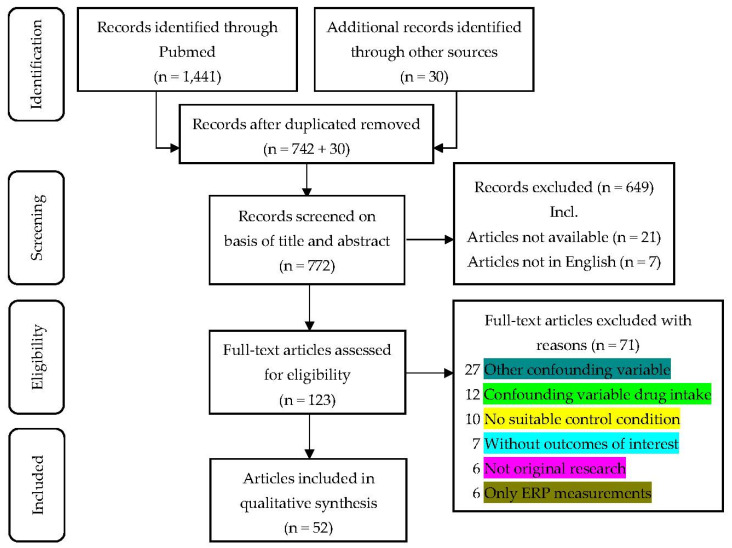
Flow diagram of article selection (The color coding refers to the classification and is picked up again in Table A1).

**Figure 2 brainsci-12-01736-f002:**
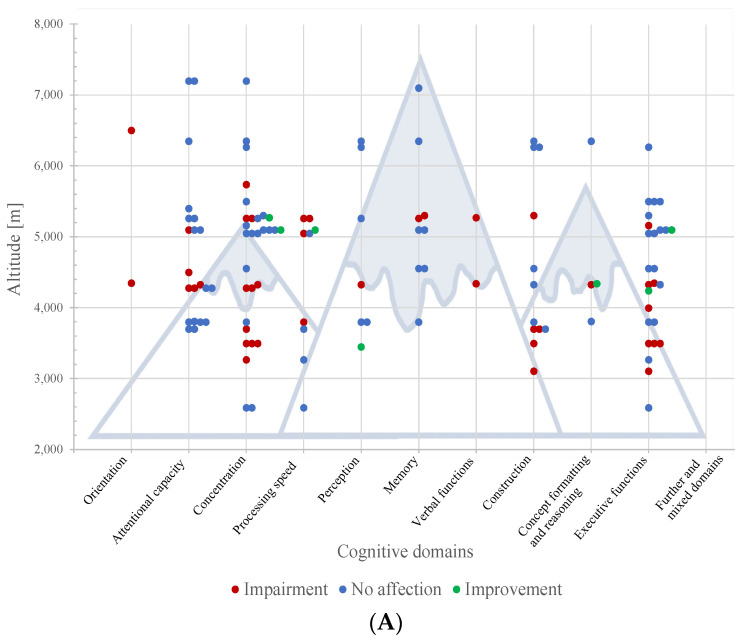
(**A**) Test results in the field studies. (**B**) Test results in the laboratory studies.

**Table 1 brainsci-12-01736-t001:** Study classification according to type of operation.

Field Studies (n = 29)		
	Passive ascent only (n = 8)	Davranche et al. (2016) [24] Frost et al. (2021) [43] Gibbons et al. (2020) [44] Latshang et al. (2013) [38] Pun, Guadagni et al. (2018) *^1^ [31] and Pun, Hartmann et al. (2018) *^1^ [32] Roach et al. (2014) *^2^ [33] and Subudhi et al. (2014) *^2^ [34] Schlaepfer, Bärtsch, and Fisch (1992) ** [37] Shi et al. (2016) [45]
	Passive and active ascent (n = 21)	Bjursten et al. (2010) [46] Bonnon, Noël-Jorand, and Therme (1999) [47] Dykiert et al. (2010) [48] Falla et al. (2021) [49] Griva et al. (2017) [50] Harris, Cleland, Collie, and McCrory (2009) [25] Issa et al. (2016) [23] Karinen and Tuomisto (2017) [40] Kramer, Coyne, and Strayer (1993) [28] Lefferts et al. (2019) [51] Lefferts et al. (2020) [52] Limmer and Platen (2018) ** [36] Malle, Ginon, and Bourrilhon (2016) [39] Merz et al. (2013) [53] Nelson et al. (1990) [54] Pelamatti, Pascotto, and Semenza (2003) [55] Petiet, Townes, Brooks, and Kramer (1988) [56] Phillips, Griswold, and Pace (1963) [57] Phillips and Pace (1966) [58] Weigle et al. (2007) [59] Zhang et al. (2013) [60]
Laboratory studies (n = 23)		
	Breathing of hypoxic gas mixture (n = 7)	Altbäcker et al. (2019) [61] Chroboczek, Kostrzewa, Micielska, Grzywacz, and Laskowski (2021) [62] Kourtidou-Papadeli et al. (2008) [41] Loprinzi et al. (2019) [63] Ochi et al. (2018) [64] Schlaepfer, Bärtsch, and Fisch (1992) ** [37] Stepanek et al. (2013) [65]
	Chamber studies (n = 16)	
	Normobaric hypoxia (n = 11)	De Aquino Lemos et al. (2012) [66] Limmer and Platen (2018) ** [36] Niedermeier et al. (2017) [67] Parker, Manley, Shand, O’Hara, and Mellor (2017) [68] Pighin, Bonini, Hadjichristidis, Schena, and Savadori (2019) [69] Pighin et al. (2012) [70] Pramsohler et al. (2017) [71] Seo et al. (2015) [72] Seo et al. (2017) [73] Turner, Barker-Collo, Connell, and Gant (2015) [74] Williams et al. (2019) [75]
	Hypobaric hypoxia (n = 5)	Abraini, Bouquet, Joulia, Nicolas, and Kriem (1998) [42] Asmaro, Mayall, and Ferguson (2013) [27] De Bels et al. (2019) [76] Nakano et al. (2015) [77] Pavlicek et al. (2005) [78]

Legend. *^n^ Same study with two articles. ** Investigation of both conditions.

## Data Availability

Not applicable.

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
