# Peer review of "Cognition and Neuropsychological Changes at Altitude—A Systematic Review of Literature"

_brainsci, 2022, doi:10.3390/brainsci12121736_

Round 1

Reviewer 1 Report

The paper titled "Cognition and neuropsychological changes at altitude 3 – a systematic review of literature" tackles an important clinical topic, i.e. clarify what cognitive functions are more sensitive to high-altitude exposure. Authors’ systematic review is methodologically well-conducted and their conclusions are interesting from both clinical and research perspectives. The paper in its current format has merit, but some points should be addressed properly to reaching a publishable level on the Journal “Brain Sciences”.

MAJOR POINTS:

1)     While this systematic review summarizes in a good way the main results of the studies in the field, the authors made little effort to try and explain why the identified cognitive deficits (i.e. mainly in the realm of attention, concentration and executive functions) are the ones more sensitive to HA exposure. The paper would draw significant benefits from an in-depth investigation of the main etiopathogenetic hypotheses of association between cognitive deficit and their underlying cerebral metabolic alteration.

2) As HA has a well-recognized impact on sleep and mood, how do the authors could exclude that the cognitive deficits they identified as mainly influenced by HA were not actually primarily influenced by sleep deprivation and/or low mood and not by the HA exposure per se?

MINOR POINTS:

1) The table 3 is not well formatted. I would advice the authors to go through it.

2) The figure captions 2A and 2B present with a different size: please double-check it.

3) As the Tables A1 and A2 are relevant but not the focus of interest of the systematic review, I would suggest the authors to upload them as "Supplementary Materials" and not to include them in the main paper, as they are very detailed and risk further burdening the reading of the article. 

Author Response

MAJOR POINTS:
1)     While this systematic review summarizes in a good way the main results of the studies in the field, the authors made little effort to try and explain why the identified cognitive deficits (i.e. mainly in the realm of attention, concentration and executive functions) are the ones more sensitive to HA exposure. The paper would draw significant benefits from an in-depth investigation of the main etiopathogenetic hypotheses of association between cognitive deficit and their underlying cerebral metabolic alteration.

Response 1) Thank you for the valuable advice. We have added some information on the brain dependency on oxygen in the introduction and are bringing this point up again in the discussion. Attentional capacity, concentration and executive functions were the most frequently studied cognitive domains. Yet, it is not possible to draw the conclusion that they were more sensitive than others, since we found impairments across all cognitive domains.

2) As HA has a well-recognized impact on sleep and mood, how do the authors could exclude that the cognitive deficits they identified as mainly influenced by HA were not actually primarily influenced by sleep deprivation and/or low mood and not by the HA exposure per se?

Response 2) You are absolutely right that factors such as sleep and mood have been shown to influence cognitive functions. Unfortunately, it is not possible to rule out influences by the above parameters in the collected studies. We have tried to underline other confounding factors when they were reported by the authors such as in Falla, Papagano et al. (2021) who found a correlation of poor performance and poor sleep. We have also added some information concerning a decrease in sleep-related memory consolidations in the introduction with Bloch et al. (2015).

MINOR POINTS:

1) The table 3 is not well formatted. I would advice the authors to go through it.

Response 1) We totally agree and have revised the arrangement of the contents in Table 3.

2) The figure captions 2A and 2B present with a different size: please double-check it.

Response 2) figure captions have been corrected. The figures themselves have been increased again for better understanding.

3) As the Tables A1 and A2 are relevant but not the focus of interest of the systematic review, I would suggest the authors to upload them as "Supplementary Materials" and not to include them in the main paper, as they are very detailed and risk further burdening the reading of the article. 

Response 3) Thank you for your suggestions. We agree and will provide Tables A1&A2 as Table S1&S2 under Supplementary Materials

Reviewer 2 Report

General and Specific Comments to the authors are included in the attached PDF document.

Author Response

General Comments:

Extensive, exhaustive and detailed descriptive review article on the topic it deals with. I consider the strategy and design to establish the selection of the articles included in this review adequate. The text appears well exposed in different subsections and thematic paragraphs in the Methods and Results sections. The global analysis that the authors make about the selected articles (and also of some others not included) to be rich, and especially the Discussion and Conclusion sections are suggestive. However, one of the Appendix Tables is overloaded with data and, as presented, it becomes somewhat difficult to visualize at some point (at the end of the specific comments, I give some idea of how it could be solved).

Response: Thank you for your very valid suggestions which helped us to improve the article. We respond to the individual suggestions below.

Specific Comments:

Line 100-101: In the number “80.0” the decimal can be eliminated. The unit of measurement “hPA” would be more correct if it is written: hPa. Likewise, the phrase would be better expressed by removing some parentheses; I propose something like the following:

“They found low paO2 (arterial oxygen partial pressure) specially between 46,7-80 hPa (35-60 mmHg) to be the main predictor…”

Response 1. Thanks for the suggestion, it was implemented that way.

Line 208: The acronym “NH” can be eliminated, since it does not appear anywhere else in the manuscript, nor in the tables and figures. Throughout the text and especially in the tables, only the acronyms “NHC” and “HHC” frequently appear. So, I think the sentence would be better as follows:

“…establishing via inhalation of hypoxic gas mixture, eleven took place in a normobaric hypoxia chamber (NHC), and in five studies the subjects were tested in a hypobaric hypoxia chamber (HHC).”

Response 2. Thanks for the suggestion, it was implemented that way.

Line 222: There is an error in the numbering of section 3.1., and it should be: “3.2. Qualitative review of the collected studies.”

Response 3. Thank you for the correction.

Line 258: Eliminate the tab in the subsection: “Attentional capacity, processing speed and working memory

Response 4. Thank you for the correction.

Line 398-401: This text does not appear tabulated and part of it is in bold type. Does it correspond to the continuation of the previous paragraph? Or would it be a new section on “Affective flexibility”?

Response 5. This appears to be a formatting error that is not present in the original document we submitted. We added the missing headline “Further and mixed domains”.

Line 438: (3)?? (Should it be indexed on the line above?)

Response 6. This again is a formatting error that is not found in the original document we submitted.

Lines 448-449: What do the authors intend to convey with the following sentence? “…no impairment occurred about as often as impairment.”

Response 7. This is a description of frequencies; in HHC, impairments were found in half of the studies and no impairments were found in the other half.

Line 459: Being two independent parts of the same figure, I propose to narrate something similar as well as the following: “Figure 2A provide an overview of the test results in field, and figure 2B the test results in laboratory studies,…”

Response 8. Thanks for the suggestion, was implemented that way.

Line 531: I think that the name of the test should appear in italics, so that there would be consistency in the presentation of the different tests throughout the manuscript: “…the Running Memory Continuous Performance Test, but the…”

Response 9. Thank you for the correction.

Line 566. The “Lake Louise Scale” is known to many Mountain Medicine specialists, but not to other scientists. Therefore, I think it would be appropriate to specify what this scale is for, as well as add a transcendent bibliographical reference to it (e.g.: Roach et al. The 2018 Lake Louise acute mountain sickness score. High Alt Med Biol 2018;19:4-6). I propose that the sentence would be better completed like this: “…as measured by the Lake Louise Scale, which is designed for the clinical diagnosis of AMS and assessment of its severity [81].”

Response 10. Thank you very much also for this suggestions which we implemented.

Line 593: Add: “…Kourdidou-Papadeli et al. [37])…”

Response 11. You are right thanks for the correction, it was implemented that way.

Figure 1: The overall figure “1,441” (in the first box) does not correlate with the sum of what is stated in lines 170-171: “1,404 articles, 21 abstracts, 14 reviews, and two meta-analyses”. As it is written, it is understood that 1,441 articles are only those that appear in PubMed, but I think it would be worth clarifying it so that there is a concordance of figures between the figure and the text (lines 170-171), and thus future readers will avoid misinterpretation.

Response 12. The sum of 1,404, 21, 14, and 2 is 1,441 records, as indicated in the first box.

Table 1: The “Laboratory Studies” section should be framed, so that the corresponding references with the two types of “Chamber studies” are clearly seen (I suggest putting some space between those that are NHC and HHC).

Response 13. The appearance is again due to reformatting and has now been adjusted accordingly.

Table 2-Legend (lines 409-410): Appears repeated: “NA: Information not available” (and also on line 406-407).

Response 14. You are absolutely right. The errors probably occurred during layout adjustments, corrections and an alphabetical order were made.

Table 3: I suggest separating the legend that contains the meaning of the acronyms of this Table, in such a way that the results that are exposed in the second part of said Table are not mixed with the legends.

Response 15. Adjustments have been made, similar to the Legend of Table 2.

Figure 2B: I think that the numbering from 0-11 that appears in small size just along the abscissa is not necessary. I suggest that these numbers be removed.

Response 16. Unfortunately, I can't figure out which small numbers on the abscissa in Figure 2B you are referring to.

Apendix A-Table A1: The acronyms that appear (AMS, HAPE, HACE, AVPU, SpO2) are familiar to some readers, but may not be to others; so, it would be desirable to include a brief legend at the bottom of said table defining such acronyms.

Response 17. Table A1 has been shifted in the Supplementary Materials, the acronym explanations have been added in the legend.

Apendix A-Table A2: This extensive Table appears somewhat cumbersome, since it is crowded with data. If the journal’s guidelines allow it, the entire Table A2 could be built on landscape pages, in such a way that the data in each column would be separated and thus its reading would be more comfortable, facilitating the understanding of the numerous messages it contains. Another option would be to include a soft gray background color in alternate columns, so that the message contained in each column would be better differentiated, and perhaps also reduce the size of the letters somewhat. Finally, the header of said Table on page 45 should be removed, so that the entire legend (containing the definitions of the acronyms that appear throughout said Table) is shown grouped.

Response 18. We are very sorry that you perceive the Table as cumbersome and have tried to make the  contents more clear. The table was in landscape format in the submitted version and was changed during the layout adjustments. For a better overview, a color adjustment of the individual columns was made and the legend was separated as suggested.

Appendix B: In the ref. 42, I suggest adjusting the spaces in the article title so that they are not confused with the “Reason”. In the ref. 64, I suggest not be cut off by pagination.

Response 19. Adjustments have been made.

References (lines 631-801): Throughout the list of references, some titles of the articles appear with the initial letter of each word in capital letters, and others do not. Please, unify their presentation based on the standard criteria of the journal.

Response 20. Thank you for the suggestion. Article names are listed as published by their respective authors. Brain Sciences does not provide standard instructions in its template and in other articles published in Brain Sciences, the references also appear sometimes without, sometimes with initial capital letters. We have therefore stuck to the previous format.

f.e.

“References should be described as follows, depending on the type of work:

  • Journal Articles:
    Author 1, A.B.; Author 2, C.D. Title of the article.Abbreviated Journal Name YearVolume, page range.

[…]

  • Unpublished materials intended for publication:
  1. Author 1, A.B.; Author 2, C. Title of Unpublished Work (optional). Correspondence Affiliation, City, State, Country. year,status(manuscript in preparation;to be submitted).”

Round 2

Reviewer 1 Report

I read the revised version of the manuscript titled “Cognition and neuropsychological changes at altitude – a systematic review of literature” and I observed that the authors made significant efforts in replying appropriately to my previous comments and suggestions.

As far as I am concerned, the present version deserves to be published.